# Multiple Therapeutic Applications of RBM-007, an Anti-FGF2 Aptamer

**DOI:** 10.3390/cells10071617

**Published:** 2021-06-28

**Authors:** Yoshikazu Nakamura

**Affiliations:** 1Division of RNA Medical Science, Institute of Medical Science, University of Tokyo, Tokyo 108-8639, Japan; nak@ims.u-tokyo.ac.jp; 2RIBOMIC Inc., Tokyo 108-0071, Japan

**Keywords:** fibroblast growth factor 2, RNA aptamer, age-related macular degeneration, achondroplasia, lung cancer, cancer pain

## Abstract

Vascular endothelial growth factor (VEGF) plays a pivotal role in angiogenesis, but is not the only player with an angiogenic function. Fibroblast growth factor-2 (FGF2), which was discovered before VEGF, is also an angiogenic growth factor. It has been shown that FGF2 plays positive pathophysiological roles in tissue remodeling, bone health, and regeneration, such as the repair of neuronal damage, skin wound healing, joint protection, and the control of hypertension. Targeting FGF2 as a therapeutic tool in disease treatment through clinically useful inhibitors has not been developed until recently. An isolated inhibitory RNA aptamer against FGF2, named RBM-007, has followed an extensive preclinical study, with two clinical trials in phase 2 and phase 1, respectively, underway to assess the therapeutic impact in age-related macular degeneration (wet AMD) and achondroplasia (ACH), respectively. Moreover, showing broad therapeutic potential, preclinical evidence supports the use of RBM-007 in the treatment of lung cancer and cancer pain.

## 1. Introduction

In mammals, fibroblast growth factors (FGF) have 22 known members that exert important functions in regulating cell proliferation, differentiation, and migration [1,2]. Upon binding to tyrosine kinase FGF receptors FGFR1–FGFR4, FGFs activate essential signaling pathways, such as the mitogen-activated protein kinase (MAPK)/ERK and JNK pathways, that are centrally involved in angiogenesis, tissue remodeling, and regeneration, including the repair of neuronal damage, skin wound healing, joint protection, and the control of hypertension. Among the FGFs, human FGF2 is an 18-kDa non-glycosylated polypeptide composed of 146 amino acids [3].

FGF2 generally plays a positive role in bone health; however, several in vitro studies have demonstrated the adverse role of FGF2 in the progression of bone disease [4,5,6,7]. Recent advances in understanding the role of FGF2 in bone formation alternatively posit that the pharmaceutical manipulation of FGF2 signaling may be a promising approach for bone disease therapy. In the literature, a number of anti-FGF2 neutralizing monoclonal antibodies (mAbs) have been reported [8,9,10,11,12], while no anti-FGF2 mAbs have been investigated for the clinical potency.

Focusing on the therapeutic potential in inhibiting FGF2, we developed the inhibitory RNA aptamer, RBM-007 [13]. In the literature, a few aptamers (DNA & RNA-based) against FGF2 (or bFGF) were described in the 1990s, but there appeared no subsequent studies on these aptamers. Aptamers are single-stranded short oligonucleotides selected in vitro from a large random sequence library, and are applicable to therapies because of several pharmaceutical advantages, such as a medium size between small molecules and antibodies, chemical synthesis, production cost, and low antigenicity [14]. Aptamers are built on the foundation of targeted molecules using the systematic evolution of ligands by exponential enrichment (SELEX) [15,16]. The concept relies on the potential of short oligonucleotides to fold, in the presence of a target, unique tertiary structures that bind with high specificity and affinity. Our studies highlight the broad therapeutic potential of RBM-007 and the multifunctionality of FGF2 (Figure 1) in the treatments of wet AMD, achondroplasia, cancer pain and lung cancer, respectively, while providing perspectives on these therapeutic applicabilities. 

## 2. Anti-FGF2 Aptamer

RBM-007 is an anti-FGF2 aptamer composed of 37 nucleotides, whose ribose 2′ positions are modified to resist ribonucleases, in addition to being 5′-PEGylated and 3′-conjugated with an inverted dT to confer an advantageous pharmacokinetic profile [13]. RBM-007 binds strongly and specifically to FGF2 and does not cross-react with other FGF family proteins or heparin-binding proteins, blocking the interaction between human FGF2 and its receptors FGFR1 through FGFR4 [16]. The dissociation constant (K_D_) of the non-PEGylated form of RBM-007 to human FGF2 protein is 2 pM, compared to 5, 7, and 27 pM in rat, mouse, and rabbit protein, respectively, showing the high affinity of RBM-007 for different FGF2s regardless of the species difference.

## 3. Therapeutic Application in Age-Related Macular Degeneration

Age-related macular degeneration (AMD) is a leading cause of blindness in the United States and Europe, and the annual incidence of neovascular AMD (referred to as “wet” AMD) in white Americans is 3.5% per 1000 people aged ≥50 years [17]. Wet AMD is caused by choroidal neovascularization, which results in disabled central visual activity due to vascular leakage and exudation, along with subretinal hemorrhage, which are often associated with subretinal fibrosis (Figure 1). Vascular endothelial growth factor (VEGF) plays a pivotal role in the pathogenesis of wet AMD. When secreted by the retinal pigment epithelial (RPE) cells from the basal side, VEGF stimulates choroidal blood vessels and promotes the expression of VEGF receptors [18,19]. Ultimately, these signaling networks lead to formation of new blood vessels that originate from the choroid, break through Bruch’s membrane, and infiltrate the macula [20]. 

The current major therapies approved by the FDA for wet AMD are pegaptanib (Macugen^®^, Eyetech Pharmaceuticals, New York, NY, USA), ranibizumab (Lucentis^®^, Roche/Genentech, San Francisco, CA, USA), aflibercept (Eylea^®^, Regeneron Pharmaceuticals, New York, NY, USA), and bevacizumab (Avastin^®^, Roche/Genentech, San Francisco, CA, USA) [19,21]. All of them target the same molecule, VEGF. Frequent intravitreal injections of anti-VEGF drugs have been shown to be associated with great visual benefits in patients with AMD [22,23,24]. Nevertheless, studies into the limitations of these therapies are well documented. In clinical trials, one-third of all treatments for naïve (i.e., untreated) patients did not respond to anti-VEGF treatment [22,23,24]. Additionally, a large number of patients successfully treated with monthly intravitreal injections of ranibizumab relapsed with worsening of vision even after two years of therapy. Poor vision and persistent exudation are often associated with macular atrophy and submacular fibrotic scar formation regardless of different anti-VEGF treatments [25]. The reasons for scarring after anti-VEGF therapy are unknown. Clinical trials have also shown submacular fibrosis developing regardless of anti-VEGF treatment, causing further health complications in wet AMD patients [26]. Finally, patients’ intravitreal injections are an additional major barrier to treatment compliance. There remains an obvious need for novel therapies with alternative mechanisms of action compared to anti-VEGF treatment for wet AMD. 

Four decades ago, FGF2 was discovered as an angiogenic factor prior to VEGF [27] and has been implicated in the pathophysiology of both angiogenesis and fibrosis (Figure 2) [28,29]. It has been demonstrated that FGF2 stimulates the growth of vascular endothelial cells and tubular structure formation [30] in addition to promoting VEGF expression [31,32]. The angiogenic activity of FGF2 was reportedly stronger than that of VEGF in a mouse corneal micropocket assay [33,34]. However, the role of FGF2 in the progression of wet AMD has never been established.

Similarly, despite the possible engagement of FGF2 in fibroblast proliferation, the involvement of FGF2 in fibrotic scar formation in the retina has long been unknown. It is known that FGF2 induces epithelial-mesenchymal transformation (EMT) in the presence of transforming growth factor (TGF)-β isomers for organ remodeling during fibrogenesis, wherein the joint action of FGF2 and TGFβ1 induced EMT of tubular epithelial cells, leading to renal tubulointerstitial scarring [35]. It has also been reported that TGFβ2, not TGFβ1, is engaged in scarring associated with eye pathologies, including proliferative vitreoretinopathy [36,37,38,39]. However, the role of FGF2 in retinal fibrosis remains unknown.

We investigated the impact of FGF2 and RBM-007 on EMT in RPE cells in the presence of TGFβ2 [40]. The study revealed that TGFβ2 alone weakly induced EMT, while FGF2 alone failed to induce EMT in RPE cells. However, FGF2 markedly enhanced TGFβ2-induced EMT. Furthermore, the enhanced EMT phenotype by the FGF2+TGFβ2 combination treatment reverted to TGFβ2 alone upon the inhibition of FGF2 by RBM-007. Consistent with these in vitro findings, in vivo studies conducted in a rat model of laser-induced choroidal neovascularization (CNV) and fibrosis indicated that intravitreal injections of RBM-007 at intervals of two weeks resulted in a statistically significant decrease in retinal fibrosis [40]. It is worth noting that this was the first report showing the therapeutic potential of RBM-007 for the prevention of retinal fibrotic scarring. 

The potency of RBM-007 in wet AMD therapy was further investigated in animal models. In mouse and rat studies, RBM-007 was effective to inhibit FGF2-induced angiogenesis and laser-induced CNV [40]. Pharmacokinetic studies indicated high and relatively long-lasting profiles of RBM-007 in the rabbit vitreous, showing superiority over other anti-VEGF medications [40]. The anti-angiogenic and anti-scarring dual action of RBM-007 might provide a highly promising additive or alternative therapy to anti-VEGF treatments in wet AMD.

Based on these preclinical findings, we initiated a phase 1/2a clinical study (referred to as the SUSHI study) in the US on wet AMD patients [41]. The SUSHI study is a non-controlled, open label, dose-escalating study with 0.2, 1, and 2 mg/eye dosing to address the safety, tolerability, and possible bioactivity of a single intravitreal injection of RBM-007 in nine patients with refractory wet AMD. A single intravitreal injection of RBM-007 under three-dosing conditions was well tolerated. Moreover, seven out of nine subjects showed evidence of RBM-007 bioactivity, in terms of any vision gain or ≥50 μm improvement in central retinal thickness after a single dose of RBM-007 in these patients who were unresponsive to prior anti-VEGF therapy. These results suggested that FGF2 participates in the pathophysiology of wet AMD independently or in conjunction with VEGF. 

Following the completion of the SUSHI study, the phase 2 (TOFU) study was initiated in December 2019 [42]. This multicenter, active-controlled, double-masked study included approximately eighty-one subjects with wet AMD. The aim is to compare the safety, efficacy, and durability of four monthly intravitreal injections of RBM-007 monotherapy, and four monthly RBM-007 injections in combination with Eylea®, dosed every other month, as well as Eylea® monotherapy dosed every other month. The study remains ongoing with further results expected by early 2022. One might speculate that if better vision acquisition is achieved by RBM-007 monotherapy or RBM-007+Eylea® combination therapy compared to Eylea® monotherapy, RBM-007 will become a tremendous game changer in the treatment of wet AMD and other retinal diseases.

## 4. Therapeutic Application in Achondroplasia

Achondroplasia (ACH) is a rare disease which causes the leading form of dwarfism in humans, occurring in approximately one in 25,000 live births [43]. ACH is caused by gain-of-function mutations in the FGFR3 gene, which encodes the transmembrane receptor, tyrosine kinase. FGFR3 transduces the extracellular communication signals triggered by FGFs [44], through substituted transmembrane and kinase domains of FGFR3, leading to ACH. In ACH, the glycine-to-arginine mutation at position 380 (G380R) is harbored by 99% of the patients, affecting the transmembrane domain of FGFR3 [45], which activates FGFR3 signaling [46]. The complex pathophysiological phenotypes induced by FGFR3 affect chondrocyte activity in multiple ways, leading to the disruption of the growth plate architecture as well as impaired endochondral bone growth [47].

Currently, there are several experimental trials based on the targeting of FGFR3 signaling for the treatment of ACH, by using small chemical inhibitors of FGFR3 catalytic activity, or downstream biomolecule-targeting pathways of FGFR3 signaling [48,49]. However, the clinical applicability of these inhibitors in ACH is complicated by the necessity of the presumptive FGFR3 inhibitor remaining active for the entire medication period from childhood for 12–14 years. In light of the toxicity of FGFR3 therapy with existing chemical FGFR inhibitors [50], a conceptually novel approach remains to be developed. One such approach is to block a cognate FGFR3 ligand, FGF2. 

We investigated the inhibitory effects of RBM-007 on FGFR3 signaling in cartilage [51]. In cultured chondrocytes, RBM-007 was found to restore the proliferation arrest, degradation of cartilaginous extracellular matrix, and premature senescence, which are attributable to the activation of FGFR3 signaling [51]. In cartilage xenografts derived from induced pluripotent stem (iPS) cells of achondroplasia patients, RBM-007 restored aberrant chondrocyte differentiation and maturation in the growth plate cartilage. Furthermore, subcutaneous injection of RBM-007 restored defective bone growth in a mouse model of achondroplasia. Consequently, neutralization of the FGF ligand by an RNA aptamer represents a viable approach to treat skeletal dysplasia caused by ectopic FGFR3 activation.

Vosoritide is a drug developed for ACH and is currently in the final phase of clinical trials [52]. As an analogue of C-natriuretic peptide (CNP) which exploits the natural mechanism regulating definitive stature in humans, vosoritide targets increased skeletal growth via mechanistic inhibition of the FGFR3 pathway involving complicated signaling networks between FGFR3/RAS/RAF/MEK/ERK [53,54]. Compared to RBM-007 that directly inhibits FGFR3 signaling, vosoritide is an indirect inhibitor of FGFR3 signaling by activating its counter-flow. Thus, while vosoritide has a significant advantage over RBM-007 with regard to clinical application, we believe therapies conceptually different from vosoritide should be explored.

Limitations of long-term vosoritide therapy exist in the complicated regulation of involved signaling networks, raising concerns about vosoritide signaling, including the appearance of resistance during long-term treatment that leads to a reduction in the overall positive clinical outcomes. Moreover, daily subcutaneous injection of vosoritide as required in the current protocol is not easy for the treatment of young patients. We believe that RBM-007, as a direct inhibitor of FGFR3, may be used either alone or in combination with vosoritide for full restoration of skeletal growth in patients with skeletal dysplasia. To provide further evidence, clinical studies of RBM-007 for ACH treatment have been initiated [55].

## 5. Therapeutic Application in Cancer Pain

Bone cancer pain is one of the most common cancer-related pains, with underlying inflammatory, neuropathic, and tumorigenic mechanisms, and is described as a severe pain with a burning and stabbing sensation [56]. Among various factors involved in the pain mechanism, nerve growth factor (NGF) is known to be a pain mediator, which transmits peripheral pain signals to the brain in addition to modulating inflammatory and neuropathic pain conditions [56]. The blockade of NGF activity by a neutralizing antibody to NGF may lead to a considerable alleviation of both ongoing as well as movement-evoked bone cancer pain-related behaviors [56]. 

We evaluated the analgesic activity of RBM-007 in a femur bone cancer (FBC) mouse model. In this model, osteosarcoma tumor cells were injected and confined to the intramedullary space of the mouse femur of the left leg, resulting in ongoing and movement-evoked pain behaviors [16]. We found that daily administration of RBM-007 for three weeks significantly restored hind paw weight distribution in a dose-dependent manner, equivalent to morphine. Importantly, late administration of RBM-007 also significantly restored hind paw weight distribution. On analyzing allodynia by the von Frey filament test, RBM-007 was discovered to increase the withdrawal threshold of an injured paw in proportion to the administration period, equivalent to the morphine level. These findings demonstrate the strong analgesic effect of RBM-007 in cancer pain [16]. 

In our understanding of pain research, this is the first report of showing in vivo evidence for the pain-modulatory action of FGF2 in bone cancer pain. Since FGF2 is known to stimulate angiogenesis and endogenous VEGF expression [57], excess FGF2 and VEGF can synergize to induce angiogenesis and increase bone cancer progression. It is known that FGF2 stimulates NGF expression [58,59]. Therefore, it is reasonable to speculate that RBM-007 reduces endogenous NGF levels by inhibiting FGF2, resulting in analgesia. However, the analgesic action of RBM-007 in bone cancer pain cannot be explained simply by reduced NGF activity, since RBM-007 is ineffective in a rat model of postoperative pain, in which anti-NGF aptamer is effective to attenuate postoperative pain [16]. The precise pain-modulatory mechanisms remain to be investigated. However, based on these results, RBM-007 is assumed an effective actor in ameliorating chronic pain induced by FGF2 overproduction.

## 6. Therapeutic Application in Lung Cancer

FGF/FGFR is a tyrosine kinase signaling pathway that regulates multiple biological events during embryogenesis, the maintenance and repair of adult organs and tissues. The FGF/FGFR pathway also affects tumorigenesis and the development of chemoresistance in various types of cancers [60]. Dysregulation of the FGF/FGFR pathway has been noted in *non-small cell lung cancer* (NSCLC), including mutations, chromosomal translocations, and gene amplifications of various members of the FGF/FGFR pathway [61]. However, FGFR inhibitors tried in several preclinical and clinical models/trials have not been found to be effective. One potential reason is that the dependence of a lung tumor on the FGF/FGFR pathway varies among tumors with activating mutations or gene amplifications [60].

Cancer-associated fibroblasts (CAFs) are known to stimulate tumorigenesis through various mechanisms (Figure 3). To assist the role of CAFs in FGF/FGFR-dependent lung cancers, a mouse model of lung adenocarcinoma has been developed by the overexpression of FGF9 in type-2 pneumocytes [62]. In adult lungs, FGF9 expression resulted in the rapid development of multiple adenocarcinoma-like tumor nodules. This model was used to investigate the role of CAFs and the FGF/FGFR signaling pathway in maintaining lung tumors initiated by FGF9 overexpression.

Findings show that FGF2 is secreted from CAFs and contributes to tumor cell growth by stimulating synthesis of more collagen and the secretion of inflammatory cell-recruiting cytokines via CAFs expressed TGFβ, MMP7, and FGF9 [62]. Moreover, CAFs stimulated the conversion of tumor-associated macrophages (TAMs) to the tumor-supportive M2 phenotype without affecting angiogenesis. Although in vivo inhibition of FGFRs by kinase inhibitors in lung tumors resulted in significant reduction in tumor size and number, withdrawal of the inhibitor induced considerable recurrence/regrowth of FGF/FGFR-independent lung tumors. When CAFs are co-cultured with tumor cell-enriched lung epithelium or wild-type lung epithelium, treatment with RBM-007 caused reduction in the colony number and size of both tumor-driven and normal lung epithelium-driven colonies. To further address the specific role of fibroblast-expressed FGF2, lung fibroblasts isolated from a null *Fgf2^−/−^* mouse were co-cultured with tumor cell-enriched lung epithelium and CAFs. Compared to co-cultures with wild-type fibroblasts, co-culture with *Fgf2^−/−^* fibroblasts resulted in fewer and smaller colonies [62]. These findings indicate that the expression of FGF2 in CAFs contributes to tumor cell growth.

Although the FGFR inhibitor significantly blocked the tumor cells, it was not enough to completely eliminate the tumor even in the presence of RBM-007, probably due to the emergence of alternative (resistance/maintenance) mechanism(s). The present findings support the need for a combinatorial strategy to treat lung cancer or other types of cancers that are heavily depending on FGF2. 

## 7. Conclusion and Perspectives

Aptamers prefer to interact with positively charged surfaces of the target proteins due to the negatively charged nature of backbone linkages. We have shown previously that a 23-nucleotide (nt) RNA aptamer interacting with the Fc domain of human IgG1 (hFc1), which lacks positive charges on the surface, can achieve strong and specific affinity to hFc1 through multiple non-electrostatic forces, such as hydrogen bonds and hydrophobic interactions [63]. This strong affinity relies on the shape complementarity between aptamers and targets [64]. Therefore, the aptamer technology is applicable not only for positively charged molecules, but also for relatively neutral molecules. 

In our laboratory, we have raised therapeutic RNA aptamers to a variety of human proteins and forwarded some of them to preclinical or clinical studies. Although aptamers share many properties with antibodies, the aptamers exhibited some superior features, such as higher affinity, medium size between small molecules and antibodies, chemical synthesis, production cost, and low antigenicity. This review summarized the properties of one such aptamer, RBM-007, against FGF2, with pre-clinical and clinical development. Reflecting the multifunctional properties of FGF2, excess FGF2 activity is involved in disease progression. Four therapeutic applications have been described here, including the treatment of wet AMD, achondroplasia, bone cancer pain, and lung cancer—all of which show RBM-007 as an expected, overall candidate for the therapeutic treatment of diseases affected by excess FGF2. 

Regarding wet AMD, we have shown in vitro and in vivo evidence that excess FGF2 plays a vital role in wet AMD by promoting angiogenesis, and that RBM-007 blocks both choroidal neovascularization and subretinal fibrosis. Moreover, combined treatment of RBM-007 and ranibizumab (Lucentis^®^) showed a synergistic effect in preventing CNV [40] although the action mechanisms remain to be clarified. This therapeutic array is further supported by the finding that FGF receptor double-conditional knockout (*Fgfr1/2*) mice showed a marked reduction in CNV accompanied by a decrease in the FGF2 level upon laser injury [65]. Additionally, FGF2 was the only essential ligand in the in vivo models of CNV, showing that FGF2 regulates pathogenic angiogenesis via the STAT3 pathway [66]. 

The therapeutic applicability of RBM-007 for achondroplasia was first suggested in an ovariectomized (OVX) rat model, a well-established model for osteoporosis; severe disruption of the epiphyseal growth plate occurred in OVX rats and RBM-007 sharply blocked the disruption of the epiphyseal growth plate in a dose-dependent manner [16]. This finding suggests that RBM-007 might ameliorate the epiphyseal growth plate and prevent skeletal dysplasias, including achondroplasia, which is related to excessive FGFR3 activation. 

In future, a long-lasting modification of RBM-007 will offer potential for extended therapeutic duration that could be beneficial for wet AMD patients who in “real-world” clinical management receive less frequent intravitreal injections, as well as for ACH young patients who receive subcutaneous injections for the entire medication period from childhood for 12–14 years. Ongoing and further clinical studies along these lines should provide us with a novel therapy for unmet medical needs and other applications, such as to diagnostics or for immune modulation.

## Figures and Tables

**Figure 1 cells-10-01617-f001:**
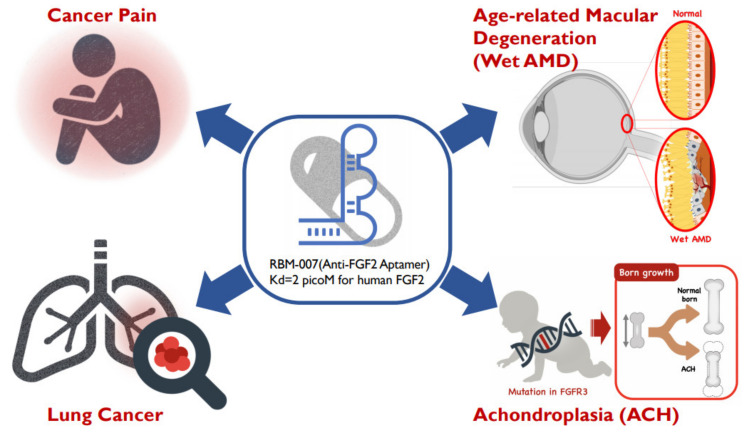
Diverse therapeutic applications of RBM-007 (anti-FGF2 aptamer).

**Figure 2 cells-10-01617-f002:**
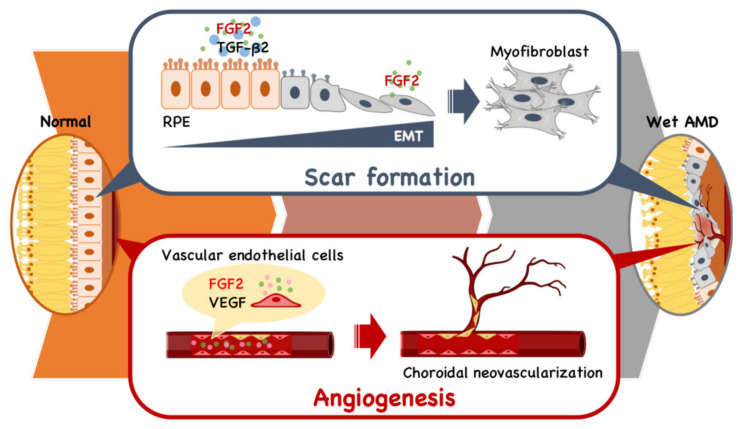
Schematic model of the dual role of FGF2 in angiogenesis and fibrotic scar formation in the retina. The presence of TGFb2 stimulates retinal pigment endothelial (RPE) cells to undergo epithelial-mesenchymal transformation (EMT) to fibroblasts, leading to scar formation. In parallel, FGF2 and VEGF act as vascular endothelial cell mitogens in the initiation and maturation of angiogenic vessels.

**Figure 3 cells-10-01617-f003:**
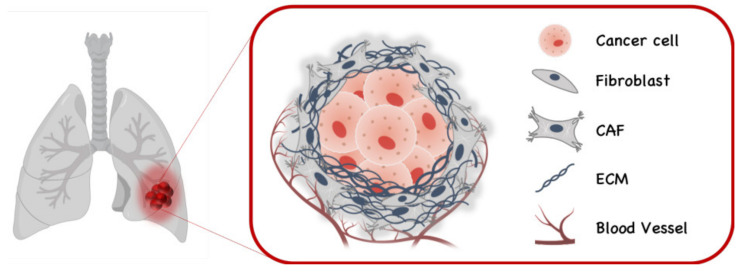
Cancer associated fibroblasts (CAFs) in lung cancer.

## Data Availability

Not applicable.

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
