# Peer review of "Multiple Therapeutic Applications of RBM-007, an Anti-FGF2 Aptamer"

_cells, 2021, doi:10.3390/cells10071617_

Round 1

Reviewer 1 Report

Dear Authors,

The manuscript entitled “Multiple Therapeutic Applications of an Anti-FGF2 Aptamer” (cells-1263771) is a comprehensively written review focusing on the therapeutical applications of RBM-007 (Anti-FGF2 Aptamer). Following are a few major points to be included in this review article.

  • In introduction section, the article explains the FGF2 growth factor, its importance and the innovative RNA aptamer (RBM-007). To further set up the story properly, please include a paragraph on other currently available anti-FGF2 aptamers (DNA & RNA-based) and the differences between other available options and RBM-007. Here, you may also specify, how these RBM-007 aptamers are superior in quality than the existing FGF2 aptamers.
  • In therapeutic applications sections (3,4,5&6), the authors discusses the in vitro and in vivo studies performed using RBM-007 for different applications. You may specify the limitations or shortcomings of RBM-007 for these individual applications and if possible, provide an outlook with the solutions.
  • In section 3 (2nd paragraph), different FDA approved anti-VEGF drugs and their limitations have been discussed. It would be nice if FDA-approved anti-VEGF aptamer (for example, Pegaptanib) based clinical study could also be included, stating if the scarring and submacular fibrosis still persisted (along with other limitations).
  • For the ease of understanding, you may also add a table, highlighting the different applications of RBM-007, with in vitro, in vivo & clinical studies, key outcomes, limitations, their references etc.
  • Also, please add a separate section on the commercialization of FGF2 aptamers, expanding on other competitive commercial anti-FGF2 aptamers already present in market with a table (advantages, limitations & status (phase1,2 or 3 trial))
  • In conclusion section, please provide an overall outlook on the current studies performed using RBM-007 aptamer, its future potential and potential applications in other biomedical areas, for example, diagnostics, etc.

For other minor comments, please refer to the attached manuscript document.

Author Response

REVIEWER #1

The manuscript entitled “Multiple Therapeutic Applications of an Anti-FGF2 Aptamer” (cells-1263771) is a comprehensively written review focusing on the therapeutical applications of RBM-007 (Anti-FGF2 Aptamer). Following are a few major points to be included in this review article.

  • In introduction section, the article explains the FGF2 growth factor, its importance and the innovative RNA aptamer (RBM-007). To further set up the story properly, please include a paragraph on other currently available anti-FGF2 aptamers (DNA & RNA-based) and the differences between other available options and RBM-007. Here, you may also specify, how these RBM-007 aptamers are superior in quality than the existing FGF2 aptamers.

Response: Thanks for this note. Regardless of the biological importance of FGF2, there is no relevant work on FGF2 aptamer that could respond to your suggestion. Hence, such a statement is added in page 1: “In the literature, a few aptamers (DNA & RNA-based) against FGF2 (or bFGF) have been described in 1990s but there appeared no subsequent studies on these aptamers.”

  • In therapeutic applications sections (3,4,5&6), the authors discusses the in vitro and in vivo studies performed using RBM-007 for different applications. You may specify the limitations or shortcomings of RBM-007 for these individual applications and if possible, provide an outlook with the solutions.

Response: The relevant descriptions are made in pages 3, 4, 6, 7 and 8 (as marked in red).

  • In section 3 (2ndparagraph), different FDA approved anti-VEGF drugs and their limitations have been discussed. It would be nice if FDA-approved anti-VEGF aptamer (for example, Pegaptanib) based clinical study could also be included, stating if the scarring and submacular fibrosis still persisted (along with other limitations).

Response: Thanks for this suggestion. The relevant description is newly made in page 2 (marked in red).

  • For the ease of understanding, you may also add a table, highlighting the different applications of RBM-007, with in vitro, in vivo & clinical studies, key outcomes, limitations, their references etc.

Response: Thanks for the suggestion. Although I found that the summary table might be reader friendly, I simply hesitate to make this because it is to repeat the statements in the text. I should appreciate your understanding.

  • Also, please add a separate section on the commercialization of FGF2 aptamers, expanding on other competitive commercial anti-FGF2 aptamers already present in market with a table (advantages, limitations & status (phase1,2 or 3 trial))

Response: At present, there is no such aptamer against FGF2 available in clinical or commercial stage.

  • In conclusion section, please provide an overall outlook on the current studies performed using RBM-007 aptamer, its future potential and potential applications in other biomedical areas, for example, diagnostics, etc.

Response: Thank you very much for this encouraging suggestion. I made two additional paragraphs in pages 7 and 8 to strengthen the current and future developments of RBM-007.

For other minor comments, please refer to the attached manuscript document.

Response: Thank you very much for numerous excellent suggestions. I revised the manuscript by incorporating all these comments. Particularly I should appreciate your suggestion of modifying the title. Also, I will ask the publisher to replace Figure 1 with high resolution.

Reviewer 2 Report

This is a well written review article, concise and understandable. The author's lab developed several therapeutic RNA aptamers to a variety of human proteins. The article describes one such aptamer, RBM-007, against FGF2, with perspectives for clinical development. Images are appropriate. References were not checked in full.

Author Response

REVIEWER #2

This is a well written review article, concise and understandable. The author's lab developed several therapeutic RNA aptamers to a variety of human proteins. The article describes one such aptamer, RBM-007, against FGF2, with perspectives for clinical development. Images are appropriate. References were not checked in full.

Response: I am very happy that you liked this review. References have been checked.

Reviewer 3 Report

In this elegant review dr Nakamura provides a nice overview of the application of the aptamer RBM-007 as a bFGF inhibitor and its functionality in multiple clinical pathologies, as treatment as of wet AMD, Achondroplasia, cancer pain and lung cancer respectively.

I really enjoyed reading this elegant overview, and my only remark would be to extend the information provided on the unique and not that well-known approach of aptamer based technology. The information provided in the current version of the manuscript is quite compact and maybe not fully clear to those reader that are not familiar with this technique (like most of the clinical readers). Expanding that section, maybe with a clarifying figure, would strengthen this interesting manuscript.

Author Response

REVIEWER #3

In this elegant review dr Nakamura provides a nice overview of the application of the aptamer RBM-007 as a bFGF inhibitor and its functionality in multiple clinical pathologies, as treatment as of wet AMD, Achondroplasia, cancer pain and lung cancer respectively.

I really enjoyed reading this elegant overview, and my only remark would be to extend the information provided on the unique and not that well-known approach of aptamer based technology. The information provided in the current version of the manuscript is quite compact and maybe not fully clear to those reader that are not familiar with this technique (like most of the clinical readers). Expanding that section, maybe with a clarifying figure, would strengthen this interesting manuscript.

Response: I am very happy that you liked this review. According to your suggestion, the extended information on aptamer in general is given in page 7.

Round 2

Reviewer 1 Report

The manuscript is sufficiently improved and authors have incorporated the reviewers suggestions (barring a few, with reasonable explanations). In my opinion, the manuscript can be accepted in the present form.